# Protocol for a realist review of General Practitioners' Role in Advancing Practice in Care Homes (GRAPE study)

Adam L Gordon ![ORCID] ,[1,2] Reena Devi,[3] Christopher Williams,[4] Claire Goodman ![ORCID] ,[5,6] Kathleen Sartain,[7] Neil H Chadborn ![ORCID] [1,2]

For numbered affiliations see end of article.

**Correspondence to**
Dr Neil H Chadborn;
Neil.Chadborn@nottingham.ac.uk

## ABSTRACT

**Introduction** Older people who live in care homes have a high level of need with complex health conditions. In addition to providing medical care to residents, general practitioners (GPs) play a role as gatekeeper for access to services, as well as leadership within healthcare provision. This review will describe how GPs were involved in initiatives to change arrangements of healthcare services in order to improve quality and experience of care.

**Methods and analysis** Following RAMESES quality and publication guidelines standards, we will proceed with realist review to develop theories of how GPs work with care home staff to bring about improvements. We identify when improvement in outcomes does not occur and why this may be the case. The first stage will include interviews with GPs to ask their views on improvement in care homes. These interviews will enable development of initial theories and give direction for the literature searches. In the second stage, we will use iterative literature searches to add depth and context to the early theories; databases will include Medline, Embase, CINAHL, PsycINFO and ASSIA. In stage 3, evidence that is judged as rigorous and relevant will be used to test the initial theories, and through the process, refine the theory statements. In the final stage, we will synthesise findings and provide recommendations for practice and policy-making. During the review, we will invite a context expert group to reflect on our findings. This group will have expertise in current trends in primary care and the care home sector both in UK and internationally.

**Ethics and dissemination** The study was approved by University of Nottingham Faculty of Medicine and Health Sciences Research Ethics Committee: 354-1907. Findings will be shared through stakeholder networks, published in National Institute for Health Research journal and submitted for peer-reviewed journal publication.

## BACKGROUND AND RATIONALE

Around 420 000 people, most of whom are over the age of 85, live in UK care homes. Care homes are primarily a social care setting and yet many residents have chronic health conditions, frailty and dementia.[1] These complex conditions can generate a diversity of care needs, which in turn require input from number of different professionals and carers.[2] All care homes rely on general practitioners

## Strengths and limitations of this study

► To our knowledge, this is the first review of how general practitioner (GP) participates in quality improvement or service development in care home settings.
► Realist review enables development of theories of how initiatives or interventions work rather than summarising outcomes data for a particular intervention.
► A potential limitation may arise from a lack of in-depth descriptions in the literature of how GPs, specifically, work with care home improvement initiatives.
► A concern is that international published evidence may not directly apply to UK context; however, the review team and context expert group are aware of differences in health systems and contexts and therefore able to comment on where evidence can inform theory development.

(GPs) to coordinate and deliver medical care and to access specialty community and hospital services for their residents. How GPs work with care homes is variable and is determined by local custom and practice, as well as the availability of other healthcare services to augment, or replace some aspects, of the GP role. Previous research has suggested that the variability in provision can result, at times, in poor care delivery, poor health outcomes for residents and increased unscheduled use of healthcare (National Health Service (NHS)) resources.[3] Within international literature, we may draw on evidence from professional roles that are equivalent to the GP-care home role, including the medical director for nursing homes in USA and elderly care physicians in the Netherlands.

Parts of the British Medical Association have suggested that it is not sustainable for GPs to continue to support the complex needs of care home residents in addition to their other work.[4] Some initiatives have sought to remove part of the responsibility for routine healthcare provision to care homes

from GPs, while others have sought to encourage GPs to become more engaged with care homes via specific commissioning arrangements and incentive payments.[5 6] It is not clear how each of these approaches influences the role, and contribution, of GPs to healthcare provision in care homes, and to the organisation, development and improvement of such provision.

The Optimal Study[7 8] identified that healthcare services for care homes achieved better outcomes when NHS staff were given time and space to develop relationships with care home staff and when their work with care homes was legitimised through role specification and recognised by their commissioning organisation/provider. Specific expertise in care of older people, particularly in the management of dementia, supported these relationships with care home staff. A further enabling feature was where multiple services were commissioned to work together and link with care home staff. This provided wrap around' support for care homes that was not reliant on single practitioners such as the GP as the main clinician. Interactions with GPs were, however, identified as being integral to how residents interpreted the quality of their healthcare, particularly around medication management and the role that the GP played in this. The study reported that the way services were organised around and with GPs could influence the willingness of GPs to engage and be proactive with care homes and their residents.

The Proactive Health Care for Older People living in Care Homes (PEACH) study[9] looked at how a quality improvement collaborative could be used to improve healthcare for older people living in care homes. It found that GPs could play a role in broader improvement initiatives, that extended beyond their specific duty of care as doctors, because they were powerful and well-connected within local health and social care economies. However, when GPs sought to play a central role, their limited capacity due to conflicting commitments could limit progress.

In many countries, developing and improving care in long-term care institutions are not the responsibility of generalist medical practitioners. In the USA, medical directors have specific obligations to support the quality of healthcare delivery in nursing homes. They undergo specific training in leadership and management competencies to support their role in service development and quality improvement.[10 11] In the Netherlands, the specialty of elderly care medicine is separate from geriatric medicine and is a primary care specialty based in nursing homes. Across different jurisdictions, doctors are expected to play an explicit role in institutional leadership with a focus on quality assurance and improvement, in addition to their specialist clinical input.[12] In initial searches, we will use the terms general practitioner and GP, and for broader searches within international literature we will use terms such as medical director, elderly care physician or primary care physician.

The NHS England Framework for Enhanced Healthcare in Care Home (EHCH), published in 2016, was proposed as the basis for a national improvement programme around healthcare in care homes.[13] EHCH laid out an approach to healthcare in care homes that favoured enhanced primary care support, access to multi-disciplinary services, access to rehabilitation, high-quality end of life and dementia care, workforce development, collaborative approaches to commissioning health and social care, and effective use of data. NHS England has stated the ambition to have every area in England develop a plan to implement the EHCH model by 2024.[14] Early evaluation of pilot sites using this approach has demonstrated better resident outcomes when compared with sites without this kind of approach.[15 16] If improvements of this kind are to be delivered at the envisaged scale and pace, then we need to understand how services can be developed, implemented and improved in care homes, and where GP engagement or support is an important requirement.

This study will develop a theory based on UK and international literature about the ways in which GPs, or other primary care doctors, have contributed to healthcare development and improvement in the care home sector. It will consider whether, and under what circumstances, GP involvement is necessary for implementation and maintenance of initiatives to improve residents' healthcare. It will explore the optimal circumstances for GPs to work effectively with other health and social care professionals to develop, implement and improve care. We anticipate that restricting literature searches to UK will result in too few articles to enable thorough development and rigorous testing of realist theory. We will therefore draw on international literature with an iterative approach; starting with initial theory development within UK literature, and broadening the search to identify whether this evidence suggests that our theories may apply to doctors and improvement initiatives in other countries, bearing in mind the different contexts; both of health systems and cultures.

The review aims to develop a programme theory of how and when GP involvement is pivotal to service development and quality improvement in care homes, and what needs to be in place to facilitate GP involvement in improving quality of care.

## PATIENT AND PUBLIC INVOLVEMENT (PPI)

To ensure that our research topic and approach are consistent with the experience and practice of care in care homes, we have consulted with the Dementia and Frail Older Persons PPI Group, Division of Rehabilitation and Ageing, University of Nottingham. We have involved one member in our project team. They suggested broadening of the research question, which initially focused specifically around GP engagement with quality improvement, to include how GPs engage with service delivery and how they interact with other healthcare professionals. They also noted that the ability of GPs to interact with other professional groups seemed integral to their general

effectiveness in working with care homes and suggested broadening the focus of the review to take account of these issues. Additional consultation with care home staff, resident and family groups will take place through the course of the project.

## RESEARCH QUESTION

How, when and under what circumstances does GP involvement in service development, implementation and improvement in care homes result in effective implementation or improved outcomes for residents?

### Aim

To understand the roles which GPs have played in the development, implementation of evidence and improvement of healthcare in care homes.

### Objectives

1. Develop a programme theory describing contexts where GPs can improve care in UK care homes, and international settings similar to UK care homes.
2. Describe the causative mechanisms whereby GP involvement in care homes results in outcomes of service development, implementation of evidence and improved quality of care.

## METHODS

Realist review is an interpretive theory-driven approach to evidence review often used to address complex issues of health service delivery. Realist approaches recognise that context always influences a programme's outcomes. By testing different plausible explanations of how particular contexts trigger responses or mechanisms to achieve (or not) certain outcomes, it provides an evidence-based narrative of what is most likely to work, how and when.[17 18] Realist theories are often expressed as a statement of: (1) context—social and environmental factors, (2) mechanism—the causal powers which lead to patterns of behaviour or choices and (3) outcome—the change in process, relationships or empirical measure.[19] Here, the social programme that we are describing relates to the role played by GPs, or primary care doctors, in service development, implementation of evidence and improvement in care homes.

This review will conform to the RAMESES quality standards for realist reviews[20 21] and will follow the outline of necessary processes as set out by Pawson.[17] We will progressively focus our review as our understanding of the topic increases. Our scope is purposively broad, in order to explore how GPs engage with a range of improvement approaches and topic areas, and the first stages of the review will iteratively develop a focus on themes which may be cross cutting.

The protocol has been registered with the International Prospective Register of Systematic Reviews.[22] It will take a four-step approach:
1. Step 1: locate existing theories.
2. Step 2: search for evidence.
3. Step 3: extracting and organising data.
4. Step 4: synthesising the evidence and drawing conclusions.

### Step 1: locate existing theories

This initial step will explore what has worked well when GPs work with care homes, how the different elements of GP working are thought to have made this happen and what needed to be in place for it to occur. The scope will include service development, delivery and improvement in care homes. This will include theories developed within Optimal[7 8] and PEACH[9] studies, as well as a wider relevant literature on the gatekeeping and leadership roles which GPs play for care homes, how care home work competes with other priorities and the relationships with other professional groups.

#### Interviews with GP leaders and practitioners

To capture the range of approaches to GP working in care homes and different theories of what is thought to work, we will conduct interviews with GPs who have been involved in healthcare improvement work in care homes or have senior leadership roles in the profession of general practice. We will recruit GPs from different parts of the UK. Within interviews, we will explore starting assumptions and what constituted success for each of these programmes, how success was achieved. We will explore the extent to which the achievement of improvement objectives was influenced by the support and involvement of GPs, and the ways in which this operated through engagement with other professional groups. We will explore how the GP contribution was affected by the presence or absence of other care professionals. Furthermore, we will explore aspects of context which acted as moderators and may be expressed as barriers to success. From these discussions, we will build an initial programme theory to test in the evidence reviewed.

#### Context expert group

We will recruit a context expert group to discuss developing theories emerging from the interviews and studies to establish if they resonate with current experience and the sociopolitical and environmental context of care homes. The context expert group will be based in UK, but we anticipate that some of the membership will have collaborations with international colleagues with relevant learning from overseas. The group will comprise 5–8 practitioners with relevant expertise on: how medical care is delivered to care homes, how new healthcare services are developed and implemented in care homes and how quality improvement around healthcare in the care home sector is undertaken. The group will comprise a mix of general practitioners: care home staff, other healthcare professionals and relatives of people who are or have received care in care homes. The group will help to refine further the initial programme theory developed through

expert consultation and subsequent iterations over the course of the study.

## Step 2: searching for evidence

Realist review is driven by an underlying logic of analysis that is designed to increase understanding and generate explanations about a topic area; thus, we will use our initial programme theory (from the literature and context expert group), to structure the evidence review. Our search strategy will be purposive and iterative with additional searching being guided by the need to find more evidence to enable us to refine our initial rough programme theory. As our programme theory becomes more refined, we may need to augment the literature with further searches to address important contextual, mechanisms or outcomes which have emerged. We will focus our initial search on documents published since 2000 and up to October 2019, since we know the bulk of published literature on service delivery in care homes has been generated in this time and because changes to service models over time, particularly with regard to GP contractual specification, will limit the usefulness of earlier publications.

Our initial search will use bibliographic databases from Medline, Embase, CINAHL, PsycINFO and ASSIA (see online supplementary appendix, eg, initial search strategy for Medline). We will consult the international literature, using terms of equivalency[23 24] to identify care home equivalents in other countries. We will seek papers describing initiatives within or applicable to the care home setting. Thus, our inclusion criteria will be: (1) designing and implementing healthcare improvements; (2) the role of medical practitioners, either in isolation or as part of a multidisciplinary team; (3) specific quality improvement. As these types of initiatives are frequently discussed outside the academic literature, we will identify grey literature through context expert group. These will include published guidelines, policy and service reports, conference proceedings and websites. Exclusion criteria will be: (1) settings for people of ages younger than 65, (2) temporary or respite stays in care homes, (3) care home admission, or GP attendance as an outcome of a study in another setting, and (4) reported in a language other than English.

Citations from the search will be selected for inclusion based on relevance and rigour.[21] Relevance relates to whether data within a document can contribute to theory building and/or testing, and rigour is whether the methods used to generate the relevant data are credible and trustworthy. A master database of the search results will be created by amalgamation of all the citations from the databases searched.

## Step 3: extracting and organising data

The initial programme theory will inform the design of a bespoke data extraction tool. During the review, we will move iteratively between analysis of particular examples of how GPs work in care homes to improve and implement changes to service delivery. At key stages, this will be shared and tested with the context expert group.

Our current knowledge of the topic area indicates that we will need to focus on and refine aspects of the review question as we go along. For example, this may include issues such as the extent to which GP involvement is a clinical consideration (medical assessment can only happen with the input of a medical practitioner) and the extent to which it reflects the broader role of the GP as a leader of primary care provision. The extent to which insights can be generalised to the UK from primary care doctors supporting care homes using different service configurations in other countries will only become clear as contextual factors and the mechanisms they trigger are identified.

Evidence reviewed will include a description of the involvement of GPs (or equivalent primary care doctors providing support to care homes in other countries), in the implementation of a new service or service model, or which describe an intervention to improve the quality of existing healthcare provision. Articles will be excluded where they describe routine healthcare provision outside the context of service development, implementation or improvement, where they describe primarily social care, or where the role of GPs is not explicitly considered. We will adopt a broad and inclusive approach to the terms implementation' and 'improvement' and what is regarded as effectiveness. We recognise that these terms are often used imprecisely by practitioners, without regard to specific theoretical frameworks from the improvement and implementation literature. In addition, we recognise that some practitioners use process measures, rather than outcome measures as evidence of effectiveness. We think it is important to capture the variation in the approaches used and how GPs influence this through their involvement. Screening at all stages of the inclusion/exclusion steps will be conducted by one reviewer. Relevant data from studies will be extracted onto a bespoke data extraction form. The list of included/excluded articles, the text of included articles and how these have been used to populate the data extraction form will be reviewed and discussed by all team members at monthly project meetings.

## Step 4: synthesising the evidence and drawing conclusions

In step 4, analysis will focus on how the evidence builds on, refutes or provides alternative explanations for, key aspects of GP's work in care homes, where outcomes may be at the level of the organisation or the resident.

Analysis will be an iterative process of proposing from the evidence different patterns of association (demi-regularities) to develop possible context–mechanism–outcome configurations that can build a theory of GP working in care homes. This is an iterative process between synthesis and analysis, refinement of the overarching programme theory and (if necessary) further iterative searching for data to test particular theories. For example, we anticipate evidence from the USA indicates that medical input requires

specialist gerontological knowledge to achieve improved resident outcomes. However, this is in the context of the US long-term care sector where GPs do not routinely support nursing homes. To understand the importance of this context to UK care homes, we would seek evidence from North American, other international and UK-derived literature, to understand whether, it is, for example, the presence of a medical physician that is important and or other professionals working with the physician, or whether there is specific specialist expertise that makes the difference.

## Outputs and dissemination

At the end of this project, we will publish a report based on our programme theory, giving practical recommendations for teams developing and improving healthcare services in care homes which will include:

1. Advice to general practice giving descriptions of the ways in which doctors can help facilitate or lead improvements in healthcare in care homes, and the ways in which interprofessional relationships contribute to outcomes.
2. Advice to care home sector staff giving practical advice as to how to engage with GPs for quality improvement in the sector.
3. Comments about resource implications for different organisations.

We will circulate these findings through academic publications in peer-reviewed journals, weblogs for stakeholder organisations including the British Geriatrics Society and Age UK, presentations at both professional (Royal College of General Practitioners, British Geriatrics Society, National Care Forum) and lay conferences (Relatives and Resident Association) in the field. We will present the findings at forums of the Enabling Research in Care Homes network for Nottingham and Derbyshire which are held twice yearly and attended by members of the public and care home staff. We will use links to the Building Community Resilience and Encouraging Independence theme within the National Institute for Health Research East Midlands Applied Research Collaboration to link this work to other programmes considering mechanisms of quality improvement and quality assurance in the care home sector. We will share our findings with healthcare practitioners and commissioners through the East Midlands Academic Health Sciences Network. Findings will also be shared with policy-makers at a national and international level via personal communication and international conferences.

## DISCUSSION

The proposed realist review has the potential to be the basis for future planning and discussion of how GPs are engaged in improvement initiatives in care homes. By better understanding what enables this group of healthcare professionals to work most effectively, there is potential to engage with concerns about workforce capacity, reduce waste and increase the efficiency of improvement and service development within the sector. These findings could be important at a time when NHS England is embarking on ambitious care home improvement initiatives in the context of challenging recruitment to general practice and limited funding.

The work will not only consider whether GPs can make a difference to improvement initiatives in care homes, but how they can make a difference and under what circumstances. The RAMESES recommendations[20 21] provide very clear guidelines to describe how decisions about literature inclusion were made, and how to describe in a transparent way how programme theories developed. We plan to avoid inadvertent biases by adhering to these. Another important limitation for dissemination is that realist review methods are relatively new to many commissioners, policy-makers and commissioners and findings need careful explanation if real-world impact is to be realised. However, the emphasis on context-sensitive recommendations offers broad principles that may be sensitively applied in different situations and circumstances. We will sense check our final publications with both our PPI representative and context expert group to ensure we explain them in the most straightforward sense.

## Author affiliations

[1]School of Medicine, Division of Medical Science and Graduate Entry Medicine, University of Nottingham, Nottingham, UK
[2]NIHR Applied Research Collaboration East Midlands, University of Nottingham, Nottingham, UK
[3]School of Healthcare, University of Leeds, Leeds, UK
[4]Department of Health Sciences, University of Leicester, Leicester, UK
[5]Centre for Research in Public Health and Community Care, University of Hertfordshire, Hatfield, UK
[6]NIHR Applied Research Collaboration East of England, University of Hertfordshire, Hatfield, UK
[7]Dementia and Frail Older Persons PPI Group, Division of Rehabilitation and Ageing, University of Nottingham, Nottingham, UK

**Contributors** RD and AG conceived the study. NC drafted the protocol, with CG and CW adding context from their clinical-academic backgrounds and KS adding context from lived-experience expertise. All authors discussed methods and objectives and have read and approved the final version.

**Funding** This study/project is funded by the National Institute for Health Research (NIHR; Health Services and Delivery Research; Project Reference: 127257).

**Disclaimer** The views expressed are those of the author(s) and not necessarily those of the NIHR or the Department of Health and Social Care.

**Competing interests** None declared.

**Patient and public involvement** Patients and/or the public were involved in the design, or conduct, or reporting, or dissemination plans of this research. Refer to the Methods section for further details.

**Patient consent for publication** Not required.

**Provenance and peer review** Not commissioned; externally peer reviewed.

**ORCID iDs**
Adam L Gordon http://orcid.org/0000-0003-1676-9853
Claire Goodman http://orcid.org/0000-0002-8938-4893
Neil H Chadborn http://orcid.org/0000-0003-1368-7983

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
