## [Reviewer comments · BMJ Open]

ARTICLE DETAILS

TITLE (PROVISIONAL)	Protocol for a realist review of General Practitioners' Role in Advancing Practice in Care Homes (GRAPE study)
AUTHORS	Gordon, Adam; Devi, Reena; Williams, C; Goodman, Claire; Sartain, Kathleen; Chadborn, Neil

VERSION 1 – REVIEW

REVIEWER	Rosemary McKenzie The University of Melbourne, Australia
REVIEW RETURNED	03-Jan-2020

GENERAL COMMENTS	This is a sound realist review protocol intended to answer important questions about the role of GPs in service delivery, development and improvement in the care of frail older people living in care homes. The service implementation, improvement and policy implications of the proposed study are significant. I would suggest minor elaboration of some aspects of the realist method and steps in theory development. In the GP interview process I would suggest that barriers to successful involvement of GPs in service improvements be explicitly explored, as well as exploring the context and contributing elements of successful GP contribution or leadership of service improvements. In the methods section, more explanation of the key realist concept of Context-Mechanism-Outcome Configurations and how they will contribute to an overarching synthesised theory, is required. Some additional references may assist with this, for example, Dalkin et al. Implementation Science (2015) 10:49 DOI 10.1186/s13012-015-0237-x AND/OR Lacouture et al. Implementation Science (2015) 10:153 DOI 10.1186/s13012-015-0345-7 I would suggest that the final recommendations resulting from the review may go beyond service improvement implications for care teams and facilities to incorporate policy recommendations of relevance to the NHS and government. Some minor editorial corrections are noted in the attached reviewed manuscript. The reviewer provided a marked copy with additional comments. Please contact the publisher for full details.
--

REVIEWER	Sarah L Brand University of Exeter Medical School, United Kingdom
REVIEW RETURNED	05-Feb-2020

GENERAL COMMENTS

I read your manuscript with great interest and valued the opportunity to review it for BMJ Open. I very much look forward to reading your findings.

I believe this to be a well-conceived realist review protocol that clearly follows the quality standards for realist review. The background and rationale for the realist review are clear. This realist review protocol paper describes a review that will fill a clear gap in the understanding of how GP involvement in care home service development, implementation and improvement in the UK can improve outcomes for patients and under which circumstances. I was particularly impressed with the PPI involvement and the PPI research team member contributing to the research question and broadening the scope of the review. The method of the four steps of realist review was clear and especially highlighted the all-important iterative nature of the searches/theory development. I enjoyed the name 'Context Expert Group'.

A general comment is that you describe searches of international literature in relation to the role of GPs in care homes in the UK. A short description of how the role of GP is defined or operationalised in an international context would be useful early in the protocol, given that GP is a UK-specific term and there may also be varied international nomenclature for care homes or similar. You do discuss this on Page 6 Line 42 but as this is a key consideration in an international review being tailored to a UK context earlier mention would be useful for the reader, especially for readers from non-UK countries. A related question that is not immediately clear is will you limit the searches to countries with a health care and social care system similar to the UK?

Specific comments to authors:

Page 2 Line 49. It should be made clear in the Abstract what Stage 2 is in the review (in brackets would suffice).

Page 2 Line 49: I suggest that the order of this sentence is reversed, in that the interviews will help form your initial programme theories, which will then guide your first round of literature searches. It would be useful to mention the iterative nature of the searches and theory development in the abstract if there is space.

Page 3 Line 22: it would be clearer if you refer to 'descriptions in the literature of how GPs...' here. Though this may be a limitation in the literature, realist review provides the opportunity to use other sources of evidence to test parts of your initial theory that the literature is sparse on, for example through your context expert group or further GP or other care team member interviews. Thus actually being a strength of realist review over other forms of more traditional systematic review (see next comment also).

Page 3 Lines 25 & 26: again, this may be a limitation in the literature (and therefore in traditional systematic review), but you would be able to use your expert group and interviews with relevant stakeholders at timely points in iterative theory development to tailor your programme theory as it develops specifically to a UK context to some degree. This can mitigate this limitation in the literature to some degree. So this and the above

	point are actually maybe better framed as reasons for using realist review rather than 'traditional' systematic review, rather than limitations of realist review, as realist review can actually mitigate these limitations in the literature compared to other forms of systematic review. Page 4 Line 22: In your research question, I was unclear what you meant by effective implementation. Firstly, what does effective mean in this context, but also, implementation of what in particular? Your research question might be clearer and easier to answer if you divided it in to a few smaller questions. In particular, consider separating delivery from implementation from improvement which may help to frame your programme theory in relation to three different but inter-related intermediate outcomes (i.e. successful delivery or implementation or improvement of service) where your overall outcome (I assume?) would be improved outcomes for residents? Further clarity around outcomes would sharpen the focus of the protocol. Page 4 Line 53: It would be useful to expand on your (useful) comment here about progressively focusing your review as your understanding increases. This will be key to successful delivery of this review which has a large topic area. I note that you discuss this with good clear examples under Step 3 and Step 4. Given the importance of this refining and focusing in a review with such a wide scope it might be worth providing more detail at this first mention of how you intend to manage the breadth of the review, and referring the reader forward on Page 4 Line 53 to where you later discuss this point in more detail. Page 5 Line 57: This first sentence is unfinished or perhaps is meant to link to the next sentence?
--	--

VERSION 1 – AUTHOR RESPONSE

Reviewer: 1

Reviewer Name: Rosemary McKenzie

Institution and Country: The University of Melbourne, Australia Please state any competing interests or state 'None declared': None declared

This is a sound realist review protocol intended to answer important questions about the role of GPs in service delivery, development and improvement in the care of frail older people living in care homes. The service implementation, improvement and policy implications of the proposed study are significant. I would suggest minor elaboration of some aspects of the realist method and steps in theory development.

Amended

In the GP interview process I would suggest that barriers to successful involvement of GPs in service improvements be explicitly explored, as well as exploring the context and contributing elements of successful GP contribution or leadership of service improvements.

Amended

In the methods section, more explanation of the key realist concept of Context-Mechanism-Outcome Configurations and how they will contribute to an overarching synthesised theory, is required. Some additional references may assist with this, for example, Dalkin et al. Implementation Science (2015) 10:49 DOI 10.1186/s13012-015-0237-x AND/OR Lacouture et al. Implementation Science (2015) 10:153 DOI 10.1186/s13012-015-0345-7

Thanks for suggested references. This section has been amended.

I would suggest that the final recommendations resulting from the review may go beyond service improvement implications for care teams and facilities to incorporate policy recommendations of relevance to the NHS and government.

Policy implications have been added.
Some minor editorial corrections are noted in the attached reviewed manuscript.
All noted with thanks.

Reviewer: 2

Reviewer Name: Sarah L Brand

Institution and Country: University of Exeter Medical School, United Kingdom Please state any competing interests or state 'None declared': None declared

I read your manuscript with great interest and valued the opportunity to review it for BMJ Open. I very much look forward to reading your findings.

I believe this to be a well-conceived realist review protocol that clearly follows the quality standards for realist review. The background and rationale for the realist review are clear. This realist review protocol paper describes a review that will fill a clear gap in the understanding of how GP involvement in care home service development, implementation and improvement in the UK can improve outcomes for patients and under which circumstances. I was particularly impressed with the PPI involvement and the PPI research team member contributing to the research question and broadening the scope of the review. The method of the four steps of realist review was clear and especially highlighted the all-important iterative nature of the searches/theory development. I enjoyed the name 'Context Expert Group'.

A general comment is that you describe searches of international literature in relation to the role of GPs in care homes in the UK. A short description of how the role of GP is defined or operationalised in an international context would be useful early in the protocol, given that GP is a UK-specific term and there may also be varied international nomenclature for care homes or similar. You do discuss this on Page 6 Line 42 but as this is a key consideration in an international review being tailored to a UK context earlier mention would be useful for the reader, especially for readers from non-UK countries.

This has been amended

A related question that is not immediately clear is will you limit the searches to countries with a health care and social care system similar to the UK?

This has been amended.

Specific comments to authors:

Page 2 Line 49. It should be made clear in the Abstract what Stage 2 is in the review (in brackets would suffice).

This has been amended.

Page 2 Line 49: I suggest that the order of this sentence is reversed, in that the interviews will help form your initial programme theories, which will then guide your first round of literature searches. It would be useful to mention the iterative nature of the searches and theory development in the abstract if there is space.

Thank you for this helpful comment, we have amended this to improve clarity.

Page 3 Line 22: it would be clearer if you refer to 'descriptions in the literature of how GPs...' here. Though this may be a limitation in the literature, realist review provides the opportunity to use other sources of evidence to test parts of your initial theory that the literature is sparse on, for example through your context expert group or further GP or other care team member interviews. Thus actually being a strength of realist review over other forms of more traditional systematic review (see next comment also).

This has been amended

Page 3 Lines 25 & 26: again, this may be a limitation in the literature (and therefore in traditional systematic review), but you would be able to use your expert group and interviews with relevant stakeholders at timely points in iterative theory development to tailor your programme theory as it develops specifically to a UK context to some degree. This can mitigate this limitation in the literature to some degree. So this and the above point are actually maybe better framed as reasons for using realist review rather than 'traditional' systematic review, rather than limitations of realist review, as realist review can actually mitigate these limitations in the literature compared to other forms of systematic review.

We agree that it is important to emphasise the strengths of realist review, and we have amended this statement.

Page 4 Line 22: In your research question, I was unclear what you meant by effective implementation. Firstly, what does effective mean in this context, but also, implementation of what in particular? Your research question might be clearer and easier to answer if you divided it in to a few smaller questions.

In particular, consider separating delivery from implementation from improvement which may help to frame your programme theory in relation to three different but inter-related intermediate outcomes (i.e. successful delivery or implementation or improvement of service) where your overall outcome (I assume?) would be improved outcomes for residents? Further clarity around outcomes would sharpen the focus of the protocol.

This is a challenging area for our review. We do not wish to be too prescriptive. We recognise from our previous work with this literature that practitioners involved in improvement may have quite different and divergent ideas of what constitutes effectiveness. For some this may be about evidencing a change in process, without extending to an evaluation of resident/patient level outcomes, whereas for others it may be about resident/patient outcome measurement without any attention to the processes that underpin these. We think that it is important to capture the range of what practitioners mean by “effective” and to comment on this – this is important to our question. and this is a contentious issue because different academics (and maybe stakeholders) have different conceptions (definitions) of what constitutes effectiveness. For example effectiveness from a care home perspective could relate to how accessible a GP service is, whereas for a commissioner and GP it could be about best use of limited resources., Similarly, we have a sense from our previous work that the terms “implementation” and “improvement” may be used imprecisely in this literature. We think it is important to capture how these terms are used, and how GP involvement relates to adoption of specified improvement approaches (such as IHI model for improvement) or implementation frameworks.

Page 4 Line 53: It would be useful to expand on your (useful) comment here about progressively focusing your review as your understanding increases. This will be key to successful delivery of this review which has a large topic area. I note that you discuss this with good clear examples under Step 3 and Step 4. Given the importance of this refining and focusing in a review with such a wide scope it might be worth providing more detail at this first mention of how you intend to manage the breadth of the review, and referring the reader forward on Page 4 Line 53 to where you later discuss this point in more detail.

Added another sentence explaining broad scope and focusing approach.

Page 5 Line 57: This first sentence is unfinished or perhaps is meant to link to the next sentence?

Apologies for this formatting error, now amended.